# What Primary Care Practitioners Need to Know about the New NICE Guideline for Myalgic Encephalomyelitis/Chronic Fatigue Syndrome in Adults

**DOI:** 10.3390/healthcare10122438

**Published:** 2022-12-02

**Authors:** Caroline Kingdon, Adam Lowe, Charles Shepherd, Luis Nacul

**Affiliations:** 1CureME, Clinical Research Department, London School of Hygiene and Tropical Medicine, London WC1E 7HT, UK; 2Centre for New Writing in the School of Arts, Languages and Cultures, University of Manchester, Oxford Rd, Manchester M13 9PL, UK; 3Myalgic Encephalomyelitis Association (MEA), 7 Apollo Office Court, Buckingham MK18 4DF, UK; 4B.C. Women’s Hospital & Health Centre, Complex Diseases Program, Vancouver, BC V6H 3N1, Canada

**Keywords:** ME/CFS, ME, understanding, diagnosis, management, disabling, stigma

## Abstract

The new NICE guideline for myalgic encephalomyelitis/chronic fatigue syndrome (ME/CFS), published in October 2021, makes significant changes in treatment recommendations. It acknowledges the complexity of this chronic medical condition, which always impacts quality of life and can be profoundly disabling, recognising the prejudice and stigma that people with ME/CFS often experience in the absence of any specific diagnostic test. The guideline outlines steps for accurate diagnosis, recognising post-exertional malaise as a core symptom; importantly, ME/CFS can now be diagnosed after just 3 months in a bid to improve long-term health outcomes. It recommends the need for individual, tailored management by a multi-disciplinary team, ensuring that the wellbeing of the individual is paramount. The guideline makes clear that any programme based on fixed incremental increases in physical activity or exercise, for example, graded exercise therapy (GET), should not be offered as a treatment for ME/CFS and emphasises that cognitive behavioural therapy (CBT) should only be offered as a supportive intervention. Because of the rigorous methodology required by NICE Committee review and the inclusion of the testimony of people with lived experience as committee members, this guideline will influence the future diagnosis and management of ME/CFS in the UK and beyond.

## 1. Introduction

The 2021 NICE guideline https://www.nice.org.uk/guidance/ng206 (accessed on 1 November 2022) aims to improve the understanding, diagnosis, and treatment of myalgic encephalomyelitis/chronic fatigue syndrome (ME/CFS). Despite recent advances, clinical judgement is critical to diagnosis because there is no definitive diagnostic test to date [1,2]. While acknowledging the lack of effective evidence-based treatments, the guideline offers guidance on care and symptom management and removes graded exercise therapy (GET) [3] as a treatment for the condition. It also downgrades the primacy of cognitive behavioural therapy (CBT) [4,5] as a treatment for ME/CFS to an intervention to support people through their chronic illness.

The new guideline acknowledges that people with ME/CFS may have experienced prejudice and disbelief [6,7] by people who do not understand their illness and encourages practitioners to consider:how this may have affected the person with ME/CFS, andthat the individual may have lost trust in health and social services and be hesitant about their involvement.

The guideline recognises that ME/CFS can cause profound, long-term disability, worsened when family, carers, employers, and clinicians fail properly to recognise the condition and its impact [8]. This has been compounded by a lack of effective treatments, wide variation in access to services, and no central register of harms experienced by patients from the treatments offered, which has served only to further alienate many people with ME/CFS and, in some cases, to undermine the confidence of those caring for them (ng206 principles of care for people with mecfs).

Changes and additions to the new NICE guideline [summarised in Box 1] include:

Post-exertional malaise is recognised as a core symptom and should be present for an ME/CFS diagnosis (ng206 nonpharmacological management of mecfs).

ME/CFS can now be confirmed in 3 months from the start of symptoms (rather than 4–6).

To balance the need for earlier diagnosis with the need for accurate diagnosis, investigations are recommended at baseline to exclude alternative diagnoses and at regular follow up to capture new or missed diagnoses.

Earlier suspicion of ME/CFS and the recommendation for initial interventions including advice on activity, energy, and symptom management aim to reduce morbidity (as a secondary prevention strategy).

People should then be directed to a specialist team experienced in ME/CFS to confirm their diagnosis and develop a personalised management plan. They should have a named contact from that team, such as a consultant physician or specialist GP, who can oversee their care and coordinate any tests and referrals.

People with ME/CFS should be advised to remain within their energy limit when undertaking activity of any kind. This means they should not push through symptoms to complete tasks, to reduce the risk of post-exertional malaise. They can be guided in this approach through energy management techniques and should balance activity with regular rest periods.

People with ME/CFS should be told about the risks and benefits of physical activity programmes and such programmes should encourage patients to remain within their energy limits. Such programmes should only be offered in selected cases and delivered by a physiotherapist in an ME/CFS specialist team.

No ME/CFS treatment programme that uses fixed or quota-based incremental increases in physical activity or exercise, including GET, should be offered to people with ME/CFS.

CBT should not be offered as a cure for ME/CFS but rather as a supportive intervention in people with ME/CFS

A greater focus is given to individualised care and support from the multidisciplinary team (MDT), and the suggested make-up of such teams has been expanded.

Box 1Summary of Messages from the 2021 ME/CFS NICE Guideline.
*Post-exertional malaise* should be present for the diagnosis of ME/CFSME/CFS may *be diagnosed after 3 months* of symptoms through a process of careful clinical assessment.*Graded exercise therapy should not be offered* as a treatment for ME/CFS.People with ME/CFS *should aim to remain within their energy limit and should not push through symptoms* to complete tasks.*Cognitive behavioural therapy should only be used as an adjunct therapy* in ME/CFS.*Individualised care should be provided by an MDT* with expanded expertise and responsibilities.


## 2. ME/CFS: The Disease

### 2.1. How Common Is ME/CFS?

Estimates suggest that around 250,000 people in England and Wales have ME/CFS [9]. Some individuals with long COVID are also being diagnosed with ME/CFS, so the number might now be much higher. In a general practice of 10,000 patients, you would expect to have around 40 people with ME/CFS, of whom 25% may be severely affected [10].

### 2.2. Why Are People Affected? What Do We Know about the Cause of ME/CFS?

ME/CFS is heterogeneous in both clinical nature and causative factors. The aetiology/cause and pathogenesis/disease process of ME/CFS are not clearly defined and uncertainty surrounds these issues.

Research has, however, identified a number of factors that are involved as predisposing factors (genetic predisposition and female gender), triggering events (infection being the commonest but this can also involve vaccinations and physical trauma) and maintaining factors (immune system dysfunction, hypoactivity of the hypothalamic-pituitary-adrenal axis, autonomic nervous system dysfunction).

ME/CFS following a viral infection is the most common presentation and is the clearest pattern to diagnose. Other immune stressors, including vaccinations, may also precede the illness [11,12]. The causes of ME/CFS are unclear, but viral infections, including COVID-19, may precipitate the development of an ME/CFS-like illness [13,14]. Post-exertional malaise in the first few months after a viral infection is usually a red flag for ME/CFS.

## 3. Recommendations from the 2021 ME/CFS Guideline

### 3.1. Defining ME/CFS

The guideline recommends that clinicians should be aware that ME/CFS:affects each person differently,varies widely in severity, from mild to very severe,is a chronic medical condition affecting multiple body systems,can have a significant impact on people’s quality of life; it is a fluctuating disease in which symptoms can change in nature and severity over days, weeks or longer—ranging from being able to carry out most daily activities to severe debilitation.

### 3.2. The Diagnosis of ME/CFS

The guideline states that ME/CFS should be suspected when:the person has had all the required symptoms [see Box 2] for a minimum of 6 weeks in adults, andthe person’s ability to engage in occupational, educational, social, or personal activities is significantly reduced from pre-illness levels, andsymptoms are not explained by another condition.

As soon as ME/CFS is suspected, the practitioner should give advice to manage symptoms—especially energy management (ng206 recommendations 1.2.6, 1.3, 1.11.2–1.11.14) It is important to make it clear to the person with suspected ME/CFS that as there is currently no diagnostic test for ME/CFS, it is recognised on clinical grounds (ng206 suspecting mecfs 1.2.1). Patients should be referred to a specialist team, where the practitioner making a diagnosis of ME/CFS should be qualified to consider all differential diagnoses (ng206 suspecting mecfs 1.2.3, 1.4).

It is critical that people with ME/CFS have access to:*an early and accurate diagnosis* so they get appropriate care, and*regular monitoring and review*, particularly when their symptoms are worsening or changing (ng206 principles of care for people with mecfs 1.1.4 and ng206 review in primary care 1.15).

Flexibility in care provision according to the needs of the individual, including remote consultations, is essential (ng206 access to care and support 1.8). Home visits may be required by those severely affected, and the practitioner is unlikely to see the person with ME/CFS at their worst (ng206 access to care and support 1.8.3). Fluctuating symptoms may lead to cancelled appointments, so flexibility is crucial (ng206 access to care and support 1.8.2). This may warrant additional support from Clinical Commissioning Groups to ensure an accessible service that is fit for purpose.

A diagnosis of ME/CFS can only be given when symptoms [Box 2] have persisted for 3 months, and other conditions have been excluded (ng206 diagnosis 1.4). Other common symptoms are listed in Box 3. Patients should be referred directly to an ME/CFS specialist team to confirm and develop a care and support plan (ng206 diagnosis 1.4.3).

Box 2Major Symptoms for Suspecting ME/CFS.
*Debilitating fatigue that is worsened by activity*, which is not caused by *excessive* cognitive, physical, emotional, or social exertion, and which is not significantly relieved by rest, and*Post-exertional malaise is the worsening of some or all symptoms after* activity, which:
-is often delayed in onset by hours or days-is disproportionate to the activity -has a prolonged recovery time, lasting hours, days, weeks or longer, and*Unrefreshing* sleep or sleep disturbance (or both), which may include:
-feeling exhausted, flu-like, and stiff on waking -broken or shallow sleep, altered sleep pattern or hypersomnia, and 
*Cognitive difficulties* (sometimes described as ‘brain fog’), including difficulty speaking, problems finding words or numbers, slowed responsiveness, short-term memory problems, and difficulty concentrating or multitasking.


Box 3Other Common Symptoms of ME/CFS.
*Orthostatic intolerance* and autonomic dysfunction, including dizziness, palpitations, fainting, nausea on standing or sitting upright from a reclining position. *Temperature hypersensitivity* resulting in profuse sweating, chills, hot flushes, or feeling very cold. *Neuromuscular symptoms* including twitching and myoclonic jerks.*Flu-like symptoms*, including sore throat, tender glands, nausea, chills, or muscle aches.*Intolerance* to alcohol, or to certain foods, and chemicals such as perfumes.*Heightened sensory sensitivities*, including to light, sound, touch, taste, and smell.*Pain*, including pain on touch, myalgia, headaches, eye pain, abdominal pain, or joint pain without acute redness, swelling or effusion. 


### 3.3. Assessment of ME/CFS

The guideline recommends that the assessment should include:a comprehensive clinical historya physical examinationa psychological and social wellbeing assessment, taking into consideration the impact of the illness, andinvestigations to exclude other diagnoses as indicated by patient history and symptom complex [see Box 4].

When ME/CFS is suspected, it is important to continue with any assessments needed to exclude or identify other conditions (ng206 suspecting mecfs 1.2.3) [see Box 4].

Box 4Tests Recommended to Exclude or Identify Other Conditions.

**Investigations to exclude other diagnoses:**
urinalysis for protein, blood, and glucosefull blood counturea and electrolytesliver function, thyroid function erythrocyte sedimentation rate or plasma viscosityC-reactive proteincalcium and phosphateHbA1cserum ferritincoeliac screen creatinine kinase

**Possible additional investigations:**
Vitamin D, Vitamin B12 and folate levelsserology, if there is a history of infection9am cortisol to exclude adrenal insufficiency



### 3.4. The Management of Suspected ME/CFS

The guideline states that when ME/CFS is suspected, people should be given personalised advice about managing their symptoms, and advice (ng206 advice for people with suspected mecfs 1.3.1):

*not to use more energy than they have*—they should manage their daily activity and not push through symptoms,

*to rest and convalesce as needed* (this may require changes to their daily routine, including work, education, and other activities),

to maintain a healthy balanced diet, with adequate fluid intake.

It acknowledges that it is important to explain to people with suspected ME/CFS that their diagnosis can only be confirmed after 3 months of persistent symptoms and the completion of investigations to exclude other conditions that may account for their symptoms (ng206 advice for people with suspected mecfs 1.3.2). Reassure them that they can return for a review if they develop new or worsened symptoms, and ensure they know who to contact for advice.

### 3.5. The Management of Confirmed ME/CFS

The new ME/CFS guideline specifies that all care should recognise (ng206 principles of care for people with mecfs 1.1.3):the reality of living with ME/CFS and how symptoms could affect the individualthe need for:
○supportive, trusting, and empathetic relationships○a person-centred approach to assess people’s needs○the involvement of loved ones and carers (as appropriate) in discussions and care planning if the person with ME/CFS chooses to include them○the need for psychological, emotional, and social wellbeing.

Within the guideline, particular consideration is given to the following issues:

#### 3.5.1. Pain

Pain should be investigated and managed according to best practice. Musculo-skeletal pain, neuropathic pain and headaches are the most common presentations. Treatments for pain that include exercise are unlikely to be suitable (ng206 symptom management 1.12.12–14).

#### 3.5.2. Cognitive Difficulties (ng206 Suspecting Mecfs) [Box 2]

A major symptom of ME/CFS, cognitive difficulties can impair not only activities of daily living, but also communication with health services, and may result in delays in health care seeking or access.

#### 3.5.3. Orthostatic Intolerance (ng206 Symptom Management 1.12.9–11)

Many people with ME/CFS experience orthostatic intolerance (ng206 symptom management 1.12.9). Pharmacological and non-pharmacologic treatments may be required to manage symptoms (ng206 symptom management 1.12.10).

#### 3.5.4. Rest and Sleep (ng206 Symptom Management 1.12.1–4)

Rest should be part of any daily routine with frequency and duration appropriate to individual needs (ng206 symptom management 1.12.1). Personalised sleep management advice should include an explanation of the role and effect of sleep disturbance in ME/CFS including broken or shallow sleep, altered sleep patterns, insomnia, or hypersomnia (ng206 symptom management 1.12.2). The emphasis should be on the gradual development of regular sleep habits, considering the need for rest in the day, and balancing this against how the person is sleeping at night (ng206 symptom management 1.12.2).

Regularly review the use of rest periods and sleep management strategies as part of the person’s care and support plan, so that this can be adjusted as needed (ng206 symptom management 1.12.4). Relaxation techniques can be helpful in rest and sleep management (ng206 symptom management 1.12.1).

#### 3.5.5. Diet and Nutrition

Adequate fluid intake and a well-balanced diet are important (ng206 symptom management 1.12.19). No dietary, vitamin or supplement interventions were recommended due to insufficient evidence favouring one over any others (ng206 symptom management 1.12.24).

Practitioners should also work with the person to find ways of minimising complications caused by gastrointestinal symptoms (such as nausea, changes to appetite, swallowing difficulties, sore throat or difficulties with the energy required to buy, prepare, and eat food). This may require medication, changes to diet or adaptations around the home.

People with ME/CFS who experience nausea should be encouraged to eat regularly, taking small amounts often (ng206 symptom management 1.12.21). If people with ME/CFS are losing weight, follow a very restrictive diet, or have unexplained weight gain, they should be referred for assessment to a dietitian with a special interest in ME/CFS (ng206 symptom management 1.12.22).

Dietetic assessment from a dietitian with a special interest in ME/CFS is particularly important for people with severe or very severe ME/CFS, who may be at risk of malnutrition (ng206 care for people with severe or very severe mecfs 1.17.2). Additional support for people with severe or very severe ME/CFS may include having nourishing drinks and snacks, including food fortification; finding easier ways of eating to conserve energy, such as food with softer textures; using modified eating aids, particularly if someone has difficulty chewing or swallowing (ng206 care for people with severe or very severe mecfs 1.17.11–12); and oral nutrition support and enteral feeding if required (ng206 care for people with severe or very severe mecfs 1.17.2).

Some people with ME/CFS may be at risk of vitamin D deficiency and may need supplementation according to the general population recommendations (ng206 symptom management 1.12.23). Explain that there is not enough evidence to support other vitamin and mineral supplements as either a treatment for ME/CFS or for managing symptoms. The potential side effects of taking doses above the recommended daily amount should be explained (ng206 symptom management 1.12.24).

#### 3.5.6. Disability Aids and Appliances

People with ME/CFS may need aids and adaptations (such as a wheelchair, blue badge or stairlift) to help maintain their independence and improve their quality of life. These should be part of the care and support plan if they would benefit (ng206 access to care and support 1.8.7–9).

#### 3.5.7. Education and Employment

As with all chronic illnesses (ng206 supporting people with mecfs in work, education and training 1.9.2–3), people with ME/CFS may need support with work, education, and vocational training, with strategies including reduced or flexible hours (ng206 supporting people with mecfs in work, education and training 1.9.4).

All practitioners should be aware that people with ME/CFS:may be unable to continue with work or education,may find that going back to work, college, or vocational programmes worsens their symptoms,are eligible to access reasonable adjustments or adaptations (in line with the Equality Act 2010) to help them continue or return to work or education (ng206 supporting people with mecfs in work, education and training 1.9.2).

It is important to offer to liaise on the person’s behalf (with their informed consent) with employers, education providers and support services and to provide information on ME/CFS, being prepared to discuss the person’s agreed care and support plan and any adjustments needed, if appropriate (ng206 supporting people with mecfs in work, education and training 1.9.1).

#### 3.5.8. Relapses

People with ME/CFS should be provided with guidance on managing flare-ups and relapses (ng206managing flare-ups in symptoms and relapse 1.14.1). Although ME/CFS is a fluctuating condition, severe or sustained worsening of symptoms should lead to a clinical review to ensure there is not another explanation (ng206managing flare-ups in symptoms and relapse 1.14.6).

Provide the person with ME/CFS advice and support so that they can identify a relapse and:*adjust their energy management* to within their current energy limits to stabilise symptoms, if possible,*conserve energy by prioritising activities* that are essential for daily living over other commitments that could be delayed or delegated,resuming any other activities *only once symptoms stabilise* and the person feels able,know who to contact in case they need a *review or further support* (ng206managing flare-ups in symptoms and relapse 1.14.5).

Some clinics reported that they provide a helpline or chat function for patients in relapse to seek advice and support without the need for an in-person appointment.

## 4. Discussion

The 2021 NICE guideline is an important document that will influence the future diagnosis and management of ME/CFS in the UK and beyond [15]. Both the patient community and health professionals have awaited its publication expectantly because of the recommendations in the earlier 2007 guideline to use GET and CBT as treatments for ME/CFS—interventions which many people with ME/CFS have found ineffective and harmful [16,17,18].

Using rigorous methodology, the NICE technical team reviewed the evidence and presented this to the committee for discussion and decision-making. There was little evidence of the efficacy of any intervention and where there was evidence, it was of low or very low quality for both non-pharmacological and pharmacological interventions. Effects, when present, were modest and may be attributed to various kinds of bias or regression to the mean. For example, there was no clinical difference in outcomes including quality of life, cognitive function, psychological status, pain, and sleep quality between CBT and standard medical care or passive control arms (ng206 evidence review for the non-pharmacological management of mecfs p. 372).

Qualitative evidence, though also of low quality, consistently showed there was a potential risk of harm from GET for some patients (ng206 evidence review for the non-pharmacological management of mecfs p. 387). This was confirmed by the clinical and lived experiences of the committee and expert witnesses. There are now multiple patient surveys, spanning two decades and over 15,000 respondents, in which many more people report harms than benefits from GET [1].

Furthermore, GET was not cost-effective, and the review of clinical trials reported that the evidence of GET as an intervention was of low or very low quality (ng206 evidence review for the non-pharmacological management of mecfs p. 387). If people suffer prolonged relapses after minimal exertions of energy, prescribing programmes that encourage pushing through symptoms is unethical.

In the 2007 NICE guideline https://www.nice.org.uk/guidance/CG53 (accessed on 1 November 2022) CBT was presented as a treatment or cure for the underlying illness, not as a supportive therapy, but the evidence from clinical trials does not support this approach, nor did the direct experience of committee members.

The reasons for the low quality of pharmacological and non-pharmacological studies included risk of bias, indirectness, and imprecision. Lack of blinding of participants was an additional problem in the non-pharmacological studies when combined with subjective measures, though they would have been less problematic if objective measures had been used (or vice versa). The result was a high risk of performance bias (ng206 evidence review for the non-pharmacological management of mecfs p. 365).

While the committee acknowledged the difficulty in blinding non-pharmacological trials, it agreed it was not impossible to do so, and that the combination of subjective measures with unblinded treatments remained an important limitation to consider when interpreting the evidence (ng206 evidence review for the non-pharmacological management of mecfs p. 365). One practical upshot of the review is that clinical trials may now attempt to avoid these same flaws going forward.

It was noteworthy that the number of studies on non-pharmacological interventions was considerable, while the number of trials on pharmacological interventions was much more limited. This imbalance in research should be addressed.

There is no evidence of curative treatments for ME/CFS, but the committee acknowledged there is a clear role for health professionals to provide supportive therapies, ranging from symptomatic treatment of symptoms such as pain, sleep disturbance and gastrointestinal problems, to the provision of emotional and lifestyle support as part of multidisciplinary care within specialist ME/CFS services. Ideally, as general practitioners become better educated in the disease, referral to specialist services will not always be necessary.

On publication of the ME/CFS draft guideline, stakeholders expressed concern that the chronic pain of people with ME/CFS would be treated according to established chronic pain guidelines. The focus of the Chronic Primary Pain Guideline https://www.nice.org.uk/guidance/ng193 (accessed on 1 November 2022) is pain that persists or recurs for longer than 3 months and cannot be accounted for by another diagnosis, or where it is not the symptom of an underlying condition. As such, the recommendations are not intended to apply to people with ME/CFS.

The committee validated the need for multidisciplinary care, which will require increased resources to meet the needs of people with ME/CFS (ng206 recommendations 1.10.1). This is particularly important for those people severely affected by ME/CFS, who have diverse needs and may need to be visited at home on a regular basis (ng206 recommendations 1.17.5). We anticipate the need for a larger number of health professionals with expertise in the diagnosis, differential diagnosis, and management of ME/CFS of various severities. Key to the successful implementation of this new guideline will be primary care practitioners working within MDTs with expertise in ME/CFS. The continuing involvement of patients and health professionals will be essential in tailoring future services to meet the needs of people with ME/CFS.

Although NICE sets guidelines for the UK, its impact is often felt further afield. Scotland is also updating its Good Practice Statement based on the guideline. Crucially, NICE joins the National Academy of Medicine (NAM) and the CDC in deprecating use of GET and the operant conditioning model of ME/CFS and joins a growing international consensus that identifies PEM as the cardinal characteristic symptom. All major UK charities and clinical or research bodies for ME/CFS support the new guidelines, including the British Association of Clinicians in ME/CFS (BACME) which represents clinicians, the UK ME Research Collaborative (MERC) which represents researchers, Doctors with ME, Physios for ME and the Forward-ME group of charities.

## 5. Conclusions

The new ME/CFS Guideline acknowledges that the management of ME/CFS often requires input from multiple health professionals, both specialists and primary care providers. GPs and other primary care providers need to recognise, acknowledge, and accept the clinical and public health significance of the condition and its impact on the life of both the individual and those who care for them. On the strength of the new guideline, primary care practitioners are empowered to refer to an appropriate specialist when the diagnosis is in question.

Furthermore, the new guideline states that all patients diagnosed with ME/CFS in primary care require referral to a specialist team for multidisciplinary input and coordination of care. It sets new standards for health professionals to ensure people with ME/CFS can access the right care and support to help them manage symptoms and makes certain no treatment that may cause harm is offered. The recommended make-up of MDTs has also been expanded to provide additional expertise throughout diagnosis and management. The guideline promises to treat people with ME/CFS with compassion, ensuring safe care of the highest standard, delivered appropriately according to the needs of the individual.

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
