# Peer review of "What Primary Care Practitioners Need to Know about the New NICE Guideline for Myalgic Encephalomyelitis/Chronic Fatigue Syndrome in Adults"

_healthcare, 2022, doi:10.3390/healthcare10122438_

Round 1

Reviewer 1 Report

The manuscript was well-written. It explains the new guideline very well and put an emphasis on the recognition and support for patients experiencing ME/CFS. There are some small typos. For example, line 122, the sentence ends abruptly. In line 355 there are 2 full stops together.

Author Response

Dear Reviewer,

Thank you for taking the time and effort to review this paper. I have amended the manuscript in response to your comments regarding the typos in lines 122 and 355 (now 135 and 356) . Thank you again.

All changes to the paper are highlighted in the attachment.

Warm regards,

Caroline

Reviewer 2 Report

The manuscript entitled “What Primary Care practitioners need to know about the new NICE guideline for myalgic encephalomyelitis/chronic fatigue syndrome in adults” aims to explain the new NICE guideline for myalgic encephalomyelitis/chronic fatigue syndrome (ME/CFS), published in October 2021, and to enable the practitioners to understand better the significant changes in treatment recommendations. It is an excellent manuscript written with great attention. It is a huge contribution to this important topic. Special interest in this topic is raised during the COVID-19 outbreak. The manuscript is very well written. I have a few small comments listed below.

-“chronic medical condition, which always impacts the quality of” line 12

-“ The guideline makes clear that any programme based on a fixed line 18

-“ for example, graded exercise therapy (GET),” line 19

-“ complete tasks to reduce the risk of post-exertional malaise. They can be guided in this” line 67

-“ In general practice of 10,000 patients, you would” line 86

-“ COVID-19 may precipitate the development of an ME/CFS-like illness (13,14)” line 93

-“ should give the advice to manage” line 112

-“ mote consultations is essential (ng206 access to care and support 1.8). Home visits may” line 125

-“ those severely affected, and the practitioner is unlikely to see the person” line 126

-“ service that is fit for purpose.” Line 130

-“ The guideline recommends that the assessment should include:” line 140

-“ serology if there is a history of infection” line 150

-“ the need for psychological, emotional, and social well-being.” Line 175

-“ cognitive difficulties can impair not only the activities of” line 183

-“ was not cost-effective, and the review of clinical trials reported” line 293

Author Response

Dear Reviewer,

Thank you for your encouraging response to the manuscript; we hope it will contribute to the field.

Regarding your comments:

-“chronic medical condition, which always impacts the quality of” line 12- changed as advised.

-“ The guideline makes clear that any programme based on a fixed” line 18: changed as advised.

-“ for example, graded exercise therapy (GET),” line 19: changed as advised.

-“ complete tasks to reduce the risk of post-exertional malaise. They can be guided in this” line 67: we believe this is correct as it stands.

-“ In general practice of 10,000 patients, you would” line 86: we believe this is correct as it stands.

-“ COVID-19 may precipitate the development of an ME/CFS-like illness (13,14)” line 93: we believe this is correct as it stands.

-“ should give the advice to manage” line 112: we believe this is correct as it stands.

-“ mote consultations is essential (ng206 access to care and support 1.8). Home visits may” line 125: we believe this is correct as it stands.

-“ those severely affected, and the practitioner is unlikely to see the person” line 126: changed as advised.

-“ service that is fit for purpose.” Line 130: changed as advised.

-“ The guideline recommends that the assessment should include:” line 140: changed as advised.

-“ serology if there is a history of infection” line 150: changed as advised.

-“ the need for psychological, emotional, and social well-being.” Line 175: changed as advised.

-“ cognitive difficulties can impair not only the activities of” line 183: changed as advised.

-“ was not cost-effective, and the review of clinical trials reported” line 293: changed as advised.

Thank you again for your helpful comments. All changes are highlighted in the attached revised paper.

Warm regards,

Caroline

Reviewer 3 Report

The authors have taken on the task to review the pragmatic aspects of the new NICE guideline for ME-CFS for primary care practitioners. This is highly commendable, as the general level of knowledge leaves a great deal of room for improvement: all parties concerned with these patients are fully aware of the shortcomings of the present level of encounter, medical work-up and follow-up in the medical practice.

There are, however, some points which could be further discussed:

1) a pure simplified readout of the main points brought about by the new guideline serves a purpose but would benefit from more of a meta-level analysis, to avoid one-sided/fragmentary views. This aspect might be improved.

- 2.2 (rows 89-95) provides a very limited view on what is known or hypothesized as the cause. To introduce the disease solely as a post-viral illness risks being one-sided, if the rest are not covered in any way. The authors may well and justifiably argue that any more in-depth coverage is beyond the scope of this paper.

- although the passage 3 contains very pertinent and great points on making the diagnosis and helping with exclusion of differential diagnoses, the management section 3.5.1-3.5.3 may be too condensed and feeble to be of use to a practitioner. Sticking strictly to what the guideline says may not be the most helpful approach all the way, but it would be possible to be analytical also as to what the guideline does not indicate.

- the discussion on the quality of studies is adequate for the purpose. How do the authors analyze the significance of the main updated items, such as cutting down the interval of symptomatic time to 3 months for the diagnosis?

2) albeit that the NICE guideline is directed to the forum of the UK, the readers would benefit from a broader perspective: what does the recommended approach mean in general. This broader view could extend to details such as prevalence figures (rr 84-87), not quite dissimilar elsewhere etc.

Author Response

Dear Reviewer,

Thank you for your very helpful review. I will address the points raised:

1) a pure simplified readout of the main points brought about by the new guideline serves a purpose but would benefit from more of a meta-level analysis, to avoid one-sided/fragmentary views. This aspect might be improved.

The authors feel that this would require a huge rewrite, but also, would be contrary to the nature of what the paper sets out to do. NICE signed off the current draft; this would require significant changes and more opinion.  We suspect NICE would be averse to such changes.

  • 2.2 (rows 89-95) provides a very limited view on what is known or hypothesized as the cause. To introduce the disease solely as a post-viral illness risks being one-sided, if the rest are not covered in any way. The authors may well and justifiably argue that any more in-depth coverage is beyond the scope of this paper.

We have now clarified this by adding in several lines (below), although we continue to believe that aetiology is beyond the scope of this paper:

NICE does not consider aetiology or pathophysiology when making recommendations for diagnosis, treatment and research, so these issues are beyond the scope of this paper.

ME/CFS is heterogeneous in both clinical nature and causative factors. The aetiology/cause and pathogenesis/disease process of ME/CFS are not clearly defined and uncertainty surrounds these issues.

Research has, however, identified a number of factors involved as predisposing factors (genetic predisposition and female gender), triggering events (infection being the commonest but this can also involve vaccinations and physical trauma) and maintaining factors (immune system dysfunction, hypoactivity of the hypothalamic-pituitary-adrenal axis, autonomic nervous system dysfunction).

- although the passage 3 contains very pertinent and great points on making the diagnosis and helping with exclusion of differential diagnoses, the management section 3.5.1-3.5.3 may be too condensed and feeble to be of use to a practitioner. Sticking strictly to what the guideline says may not be the most helpful approach all the way, but it would be possible to be analytical also as to what the guideline does not indicate.

  • the discussion on the quality of studies is adequate for the purpose. How do the authors analyze the significance of the main updated items, such as cutting down the interval of symptomatic time to 3 months for the diagnosis?

We believe that we have provided adequate detail in Sections 3.5.1-3.5.3 for the purposes of this paper.

2) albeit that the NICE guideline is directed to the forum of the UK, the readers would benefit from a broader perspective: what does the recommended approach mean in general. This broader view could extend to details such as prevalence figures (rr 84-87), not quite dissimilar elsewhere etc.

We have added the following sentences to the discussion:

Although NICE sets guidelines for the UK (with the exception of Scotland), its impact is often felt further afield. Scotland is also updating its Good Practice Statement based on the guideline. Crucially, NICE joins the National Academy of Medicine (NAM) and the CDC in deprecating use of GET and the operant conditioning model of ME/CFS, and joins a growing international consensus that identifies PEM as the cardinal characteristic symptom. All major UK charities and clinical or research bodies for ME/CFS support the new guidelines.

I hope this has addressed your comments. Thank you again for your helpful review. All changes in the document are highlighted in yellow in the attached file.

Warm regards,

Caroline

Reviewer 4 Report

I read with great interest the paper written by the authors. The number of patients diagnosed with myalgic encephalomyelitis/chronic fatigue syndrome (ME/CFS) is surprisingly increasing during the COVID-19 pandemic. Therefore, general practitioners should know the management of ME/CFS with the latest guideline.

It is an excellent paper, as it comprehensively describes the NICE guidelines for ME/CFS published in 2021 and describes what general practitioners need to know, with a focus on the changes from the previous 2007 guidelines.

As one of the reviewers, I have no specific recommendations for improvement.

Author Response

Thank you for your encouraging review.

In response to the other reviewers, some suggested changes have been made, which are highlighted in the attached revised paper.

Warm regards,

Caroline
